# Ectopic expression of *Aspergillus flavus* uricase and URAT1 in therapeutic cells promotes intracellular degradation of uric acid in hyperuricemic mice

Yuzhong Feng[1], Jiazhen Cui[1], Xuan Huang[2], Yupeng Li[2], Haolong Dong[1], Xianghua Xiong[1], Gang Liu[1], Qingyang Wang[1]*, Huipeng Chen[1]*

1 Academy of Military Medical Sciences, Beijing, China, 2 Institute of Life Sciences and Medical Engineering, Anhui University, Hefei, China

* tansun0532@163.com (QW); chenhuipengbj@163.com (HC)

## Abstract

Uricase-based drugs excel at treating refractory hyperuricemia and tumor lysis syndrome by directly degrading uric acid but are limited by immunogenicity. Here, we engineered RAW264.7 macrophages with ectopic co-expression of *Aspergillus flavus* uricase and murine urate anion transporter 1 (URAT1), forming a "transport-degradation" system: URAT1 actively transports uric acid into cells for intracellular degradation. Recombinant lentiviral vectors carrying target genes were transfected into RAW264.7 cells, followed by puromycin screening. *In vitro* assays showed that the engineered macrophages nearly completely degraded uric acid (from $556.0 \pm 37.0$ µmol/L to $0.7 \pm 0.6$ µmol/L) at 72 h. URAT1 inhibition with benzbromarone abolished uric acid degradation in URAT1-expressing cells. In both acute dietary-induced and chronic genetic hyperuricemic mouse models, RAW-afUri-URAT1 exerted robust and sustained uric acid-lowering activity, maintaining serum uric acid at $77.14 \pm 37.48$ µmol/L on day 16 in yeast extract gavaged mice and normalizing serum uric acid to $76.2 \pm 15.9$ µmol/L in liver uricase conditional knockout mice, both significantly superior to the rebound levels observed in mice treated with Rasburicase ($143.19 \pm 38.21$ µmol/L and $142.4 \pm 17.4$ µmol/L, respectively; $P < 0.05$). Safety assessments in dietary-induced hyperuricemia mice showed no obvious abnormalities in liver or renal function, and significantly reduced hyperuricemia-related production of inflammatory cytokines (IL-1β, IL-6, TNF-α), Immunogenicity assays showed undetectable anti-uricase antibodies in RAW-afUri-URAT1 treated mice but high level of antibodies in rasburicase treated mice. This engineered macrophage system shows potent, durable uric acid-lowering efficacy, with low immunogenicity and good biosafety, offering a promising strategy for hyperuricemia therapy.

**Data availability statement:** All relevant data are within the manuscript and its Supporting information files.

**Funding:** The author(s) received no specific funding for this work.

**Competing interests:** The authors have declared that no competing interests exist.

## Introduction

Hyperuricemia, defined as a serum uric acid (sUA) concentration exceeding 420 µmol/L, has evolved from a neglected metabolic disorder to a major public health challenge globally. Fueled by dietary shifts toward high-purine foods, sedentary lifestyles, and aging populations, its prevalence continues to rise, with significant implications for associated complications such as gout, chronic kidney disease, cardiovascular disorders, and metabolic syndrome [1–5]. The clinical management of hyperuricemia has long relied on two classes of traditional drugs: xanthine oxidase inhibitors (e.g., allopurinol, febuxostat) that suppress uric acid (UA) production, and uricosuric agents (e.g., benzbromarone) that enhance renal UA excretion [6–8]. However, these conventional therapies face inherent limitations that compromise their efficacy and safety: xanthine oxidase inhibitors often require prolonged administration to achieve target UA levels and may induce primary or acquired resistance due to individual metabolic variability. Febuxostat carries potential cardiovascular risks and XOI use may even increase all-cause mortality in patients with heart failure [9]. Uricosuric agents, on the other hand, are contraindicated in patients with impaired renal function and carry a risk of nephrolithiasis [10]. More critically, neither class can effectively address refractory hyperuricemia—cases unresponsive to maximum doses of conventional drugs—or promote the dissolution of established gouty tophi, as they only indirectly modulate UA levels without eliminating pre-existing UA deposits [11–13]. In contrast, uricase-based drugs have emerged as a transformative therapeutic option, distinguished by their unique mechanism of action. Unlike traditional agents that act on metabolic pathways, uricase directly catalyzes the oxidation of poorly soluble UA into allantoin, a compound with approximately 5–10 times higher water solubility that is rapidly eliminated via the kidneys [14,15]. This direct degradation mechanism endows uricase with unparalleled efficacy, enabling rapid and profound reductions in sUA levels even in refractory cases and facilitating gouty tophi dissolution [16,17]. For patients with severe hyperuricemia or those intolerant to conventional drugs, uricase represents an irreplaceable clinical tool, addressing the unmet needs that have long plagued hyperuricemia treatment.

Despite their remarkable therapeutic potential, the widespread clinical application of uricase-based drugs is hindered by a critical barrier: immunogenicity. Humans lack a functional uricase gene (a result of evolutionary pseudogenization) [18,19], meaning exogenous uricase—whether derived from microbes, animals, or recombinant sources—is recognized as a foreign antigen by the immune system. This triggers a cascade of immune responses, including the production of anti-drug antibodies (ADAs), allergic reactions ranging from skin rashes to life-threatening anaphylaxis, and accelerated drug clearance, ultimately leading to diminished efficacy or complete therapeutic failure [20,21]. To overcome this limitation, the development of advanced delivery systems has become the central focus of uricase research. Current strategies include chemical modification (e.g., pegylation to shield antigenic epitopes) [22,23], encapsulation in nanocarriers (liposomes, polymeric nanoparticles, or inorganic nanomaterials) [24–26], and cell-based delivery platforms [27]. Pegylated uricase (e.g., pegloticase) has shown improved pharmacokinetics and reduced

immunogenicity compared to unmodified enzymes, but ADAs still develop in a substantial proportion of patients [28]. Nanocarriers can protect uricase from immune recognition and prolong circulation time, yet they often suffer from issues such as rapid clearance by the mononuclear phagocyte system, potential cytotoxicity, and inconsistent drug release [29]. Current cell-based delivery system, like the engineered red blood cell (Uri@RBC) [27], boasts prolonged circulation and low immunogenicity, but it still faces challenges including potential reduction in biocompatibility, low efficiency of UA uptake, and unknown immune risks..

In this study, we developed a novel, efficient, and low-immunogenicity uricase delivery system by leveraging the functional synergy between key molecules. We utilized URAT1 (encoded by *Slc22a12*), a key UA transporter predominantly localized to the apical membrane of renal proximal tubule cells and uniquely capable of mediating the specific, unidirectional uptake of extracellular UA into cells [30], to address the core limitation of passive diffusion in current uricase delivery systems. For the therapeutic payload, we chose *Aspergillus flavus* (*A. flavus*) uricase, whose recombinant form rasburicase has been globally approved for the treatment of hyperuricemia and tumor lysis syndrome with robust efficacy and a well-established safety profile [31]. Murine macrophage cell line RAW264.7 cell is selected as the delivery vector in animal experiment for it could be smoothly expanded to autologous macrophage in human therapy [32]. We engineered RAW264.7 cells to co-expressed *A. flavus* uricase and URAT1 to form a coordinated "transport-degradation" mode—URAT1 actively shuttles extracellular UA into cells, where it is efficiently degraded by intracellular uricase, a mechanism that aligns with the physiological paradigm of UA metabolism in mammals [33,34]. This engineered macrophage exhibits prominent UA-lowering effects *in vitro* and *in vivo*, and significantly reduces the immunogenicity associated with conventional uricase-based UA-lowering agents.

## Materials and methods

### Construction of recombinant lentiviral vectors

The recombinant lentiviral vectors were designed based on the GL181 lentiviral plasmid (OBiO, Shanghai, China) as the backbone, which is inherently equipped with an EGFP reporter gene and PuroR gene (puromycin resistance gene for cell selection). *A. flavus* uricase and *Slc22a12* gene (encoding URAT1) were sequentially inserted into the multiple cloning site of GL181 plasmid. Specifically, the uricase gene was FLAG-tagged at 5' terminal and cloned between the EcoRI and BamHI restriction sites, while the *Slc22a12* gene was linked to uricase via a P2A peptide sequence, forming the recombinant vector GL181-afUri-URAT1 (Fig 1A). RAW264.7 cells transfected with GL181-afUri-URAT1 express a non-secreting uricase. As contrast, a uricase-secreting RAW264.7 cell is needed as well. The vector GL181-Igκ-afUri was constructed by fusing the Ig Kappa signal peptide sequence to the 5' end of the uricase gene, followed by insertion into the same EcoRI and BamHI sites of the GL181 plasmid to enable extracellular secretion of uricase (Fig 1B). As control, a non-secretory vector GL181-afUri (short for *A. flavus* uricase in vectors and cells) was also constructed without the fusion of the Ig Kappa peptide sequence(schematic structure not shown). Empty lentiviral vector GL181 plasmid was used as negative control (Fig 1C). Recombinant lentiviruses were commercially prepared by OBiO company through standard packaging procedures in 293T cells, with the titer of the purified virus confirmed to be over $1.0 \times 10^8$ TU/mL.

### Lentiviral transfection and enrichment of engineered RAW264.7 cells

$2 \times 10^5$ cells of RAW264.7 were seeded in each well of a 6-well plates, and cultured overnight in a 5% $CO_2$ incubator. The original medium in 6-well plates was discarded when the cell density reached 30–40%. The concentrated lentivirus was added at a multiplicity of infection (MOI) of 50; meanwhile, polybrene-plus (OBiO, Shanghai) was added at a final concentration of 1 mg/mL. After 24 h of infection, the cells were refreshed with complete DMEM medium, and the cells continued to be cultured in a 5% $CO_2$ incubator. 72 h post-infection, EGFP expression was observed under a fluorescence microscope, and the positive cell rate was detected by flow cytometry. Subsequently, puromycin (final concentration 10 µg/mL) was added to the culture medium for 3 days until stably transfected engineered macrophages were enriched.

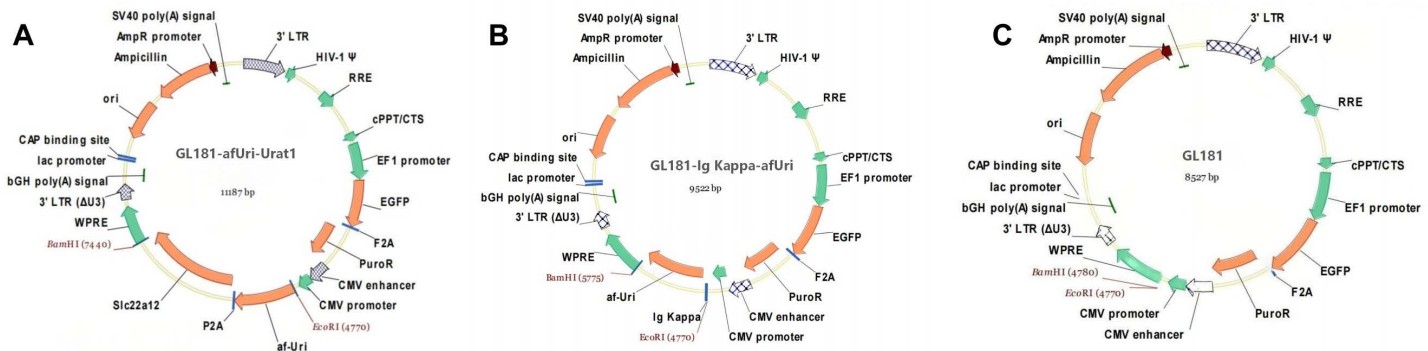

**Fig 1. Schematic structures of lentiviral transfer vectors. (A)** Modified vector with *A. flavus* uricase insertion, linked with *Slc22a12* gene via P2A peptide sequence. **(B)** Vector with Igκ-afUri insertion. **(C)** Basic lentiviral vector backbone GL181 plasmid, used as negative control.

## Verification of uricase expression

Western blotting was used to verify the expression of *A. flavus* uricase (FLAG-tagged) in engineered macrophages. Total protein was extracted from transfected cells using RIPA lysis buffer supplemented with 1% protease inhibitor cocktail (MedChemExpress, HY-K0010), followed by centrifugation at 12,000 rpm, 4°C for 15 min. The supernatant was separated by SDS-PAGE and transferred onto NC membrane. After blocking with 5% skimmed milk, the membrane was incubated with primary antibody against FLAG tag (Cell Signaling Technology, #14793, 1:1000 dilution) at 4°C overnight, followed by Horseradish Peroxidase-conjugated Goat Anti-Rabbit IgG (Jackson ImmunoResearch, 111-035-144, 1:2000 dilution) at room temperature for 1 h. The protein bands were visualized using an ECL chemiluminescence kit (Biodragon, BF06053).

To detect the secretory status of FLAG-tagged uricase, RAW264.7 cells ($1 \times 10^6$ cells/mL) were seeded into a 12-well plate and cultured for 12 h. The original medium was replaced with 400 µL of DMEM containing 1% FBS, and incubated for another 24 h. Culture supernatants were collected and centrifuged at 6000 rpm, 4°C for 5 min to remove cell debris. The subsequent SDS-PAGE separation, membrane transfer, antibody incubation, and chemiluminescence detection procedures were identical to those described above.

## *In vitro* UA-Lowering Efficacy Assay

To evaluate the *in vitro* UA-lowering efficacy of engineered cells, stably transfected RAW-afUri-URAT1, RAW-Igκ-afUri, RAW-afUri, and RAW-Ctrl cells were subjected to functional assays. Cells were seeded in 12-well plates at a density of $5 \times 10^5$ cells/well in 1 mL of complete DMEM medium, followed by the addition of 100 µL of high-concentration UA stock solution (ThermoScientific, 171291000) to establish the UA-rich culture system. Each experimental group was triplicated to ensure statistical reliability. Culture supernatants were collected at 0, 24, 48, and 72 h post-seeding, and the UA concentration in each supernatant was quantified using an automatic biochemical analyzer (BioBase, China) following the manufacturer's protocol.

To clarify the UA-lowering mechanism of each engineered cell, we further assessed the UA-lowering capacity of the cell culture supernatant. Briefly, 1 mL of cell suspension containing $5 \times 10^5$ cells was seeded into each well of a 12-well plate and cultured for 48 hours (without the addition of UA solution). Subsequently, 900 µL of cell culture supernatant was collected and mixed with 100 µL of a high-concentration UA stock solution. The mixture was incubated in a 37°C incubator, and 100 µL aliquots were sampled at 0, 24, 48, and 72 hours post-incubation, and the UA concentration was quantified using the biochemical analyzer.

### *In vitro* URAT1 inhibition assay

To confirm the dependence of UA degradation on URAT1-mediated transport, URAT1 inhibition assays were performed. RAW-afUri-URAT1 cells were seeded into 12-well plates and treated with benzbromarone (10 μM), lesinurad (10 μM), or vehicle control (DMSO, final concentration 0.001%). High-concentration UA was added to the medium, and cells were incubated for 72 h. Supernatants were collected, and UA concentrations were measured using the automatic biochemical analyzer.

### Establishment of hyperuricemic mouse models and *in vivo* efficacy evaluation

Yeast Extract-Induced KM Mouse Model: The hyperuricemia model was established and the protocols were described in our previous published work [35]. Specifically, KM mice were randomly divided into 6 groups (n = 8 per group): Mock group, Model group, RAW-Ctrl group, RAW-Igκ-afUri group, RAW-afUri-URAT1 group, and Rasburicase group. Detailed information on the groupings and key interventions for each group is summarized in S1 Table. Except for the Mock group, mice in other groups were given 15 g/kg yeast extract by gavage once a day for 16 consecutive days to establish the hyperuricemic model (according to former literature, with modification from pilot experiments) [36]. On days 1 and 9, mice in the RAW-Ctrl, RAW-afUri-URAT1, and RAW-Igκ-afUri groups received a single tail vein injection (i.v.) of the corresponding engineered macrophages at a dose of $2 \times 10^6$ cells/mouse. For the Rasburicase group, mice were administered rasburicase i.v. at a dose of 0.2 mg/kg body weight for three consecutive days starting on day 1 (days 1~3) and again starting on day 9 (days 9~11). Blood samples were collected at 0, 4, 8, 12, and 16 days, and the sUA was measured using a UA detection kit (Solarbio, China) to evaluate the *in vivo* UA-lowering effect.

Liver Uricase Conditional Knockout (Uox-CKO) Mouse Model: Liver-specific uricase knockout C57BL/6J mice were generated and identified by Sipeifu Biotechnology Co., Ltd. (Beijing, China) using the Cre-loxP system according to previously reported protocols [37]. Thirty mice were randomly divided into 6 groups (n = 5 per group): Mock (Cre⁻), Model (Cre⁺), RAW-Ctrl (Cre⁺), RAW-afUri-URAT1 (Cre⁺), RAW-Igκ-afUri (Cre⁺), and Rasburicase (Cre⁺). Engineered macrophages ($1 \times 10^6$ cells/mouse) were injected via tail vein on days 1, 8, 15, and 22. Rasburicase (0.2 mg/kg) was administered on days 1~3, 8~10, 15~17, and 22~24. Due to the limited blood volume of C57BL/6 mice, serial blood collection for dynamic monitoring of UA levels was not feasible. Instead, blood samples were collected only at the experimental endpoint (day 28) for UA measurement.

All mice were housed in a specific pathogen-free (SPF) facility with a 12-hour light/12-hour dark cycle, controlled temperature (22~24℃), and free access to food and water. For procedures like gavage and tail vein injection, mice were anesthetized by inhalation of 3–5% isoflurane delivered via a precision vaporizer in an induction chamber to alleviate procedural discomfort. At the experimental endpoint, all mice were euthanized by exposure to 100% carbon dioxide ($CO_2$) at a displacement rate of 50% of the chamber volume per minute, followed by cervical dislocation to confirm death. All efforts were made to minimize animal suffering and to use only the number of animals necessary to produce reliable scientific data.

### Serum biochemistry and inflammatory cytokines in KM mice

At the end of the KM mouse experiment, serum levels of alanine transaminase (ALT), aspartate transaminase (AST), total bilirubin (TBIL), direct bilirubin (DBIL), creatinine (CRE), and UREA were measured using the automatic biochemical analyzer. Serum concentrations of IL-1β, IL-6, and TNF-α were quantified using commercial ELISA kits (Dakewe Biotech, Shenzhen, China).

### Detection of anti-uricase antibodies

Serum anti-uricase antibodies were detected using indirect ELISA methods with a universal ELISA auxiliary kit (Dakewe Biotech, Shenzhen, China). Serum samples were obtained from four treatment groups in the liver uricase conditional

knockout (Uox-CKO) mouse experiment at day 28: RAW-Ctrl, RAW-afUri-URAT1, RAW-Igκ-afUri, and Rasuricase groups. Microtiter plates were coated with either rasburicase or secreted *A. flavus* uricase from RAW-Igκ-afUri cell supernatants, followed by blocking with 5% BSA. Serum samples were diluted 1:20 and incubated at 37℃ for 1 h. Horseradish peroxidase (HRP)-conjugated goat anti-mouse IgG secondary antibody was diluted 1:2000 and incubated at 37℃ for 1 h. The TMB substrate was used for color development, and optical density (OD) values at 450 nm were measured to represent relative antibody levels.

### Statistical analysis

All experimental data were expressed as mean ± standard deviation (SD) and analyzed using GraphPad Prism 8.0 software. Differences between groups were compared by one-way analysis of variance (ANOVA) followed by Tukey's post-hoc test. $P < 0.05$ was considered statistically significant.

## Results

### Construction and validation of engineered RAW264.7 macrophages

We employed lentiviral transduction to infect macrophages. First, we constructed recombinant lentiviral vectors: the UA transporter vector GL181-afUri-URAT1 (co-expressing URAT1 and *A. flavus* uricase), the uricase-secreting vector GL181-Igκ-afUri (expressing secretory uricase fused with an Ig-Kappa signal peptide to facilitate extracellular secretion), and the negative control vector GL181. All vectors harbored EGFP as a reporter gene and a puromycin resistance gene for positive cell selection. After transducing RAW264.7 cells, fluorescence microscopy images revealed EGFP expression in all three groups of engineered RAW264.7 cells, indicating successful lentiviral transduction and target gene expression (S1 Fig). The proportion of EGFP-positive cells was quantified by flow cytometry (S2 Fig), showing positive rates of 9.0%, 9.4%, and 20.1% in the RAW-afUri-URAT1, RAW-Igκ-afUri, and RAW-Ctrl groups, respectively. This relatively low initial transduction efficiency is consistent with the well-documented inherent resistance of macrophage cell lines to viral transduction [38,39].

As the percentage of EGFP positive cells was low, puromycin was utilized to enhance the percentage of positive cells. After three round cultures with 10 µg/mL puromycin, the percentage of EGFP positive cells rose up to 91.2%, 90.4%, and 97.0%, for RAW-afUri-URAT1, RAW-Igκ-afUri, and RAW-Ctrl, respectively, as shown by fluorescence microscope (Fig 2, A-C) and flow cytometry (S3 Fig).

To further determine whether *A. flavus* uricase was expressed in engineered RAW267.4 cells, western blot analysis was performed. As FLAG tag was fused to the uricase gene, the expected molecular weight of the FLAG-uricase fusion protein was about 34 kDa. As shown in Fig 2D, no specific bands were detected in wild-type RAW264.7 cells (RAW-WT) or the RAW-Ctrl group, which was consistent with their lack of the FLAG tagged uricase gene. The RAW-afUri group (expected to express a non-secreted uricase, described in "Materials and Methods") exhibited a single specific band that matched the expected molecular weight of FLAG-uricase (~34 kDa), confirming successful expression of the target protein. For the RAW-Igκ-afUri group (expected to express a secretory type with Igκ signal peptide), two distinct bands were observed with the main larger band indicating the Igκ signal peptide-fused uricase (~36 kDa) while the minor and smaller band indicating a cleavage of the Igκ signal peptide from the fusion protein (~34 kDa). Notably, the RAW-afUri-URAT1 group showed a single specific band at ~37 kDa indicating a P2A fragment attached to the N-terminus of uricase, resulting in a ~2–3 kDa increase in apparent molecular weight consistent with previous reports [40]. Collectively, these results confirm that FLAG-uricase fusion proteins are specifically expressed in the three engineered macrophage groups, with expression patterns (molecular weight and band number) consistent with their respective plasmid design and functional characteristics.

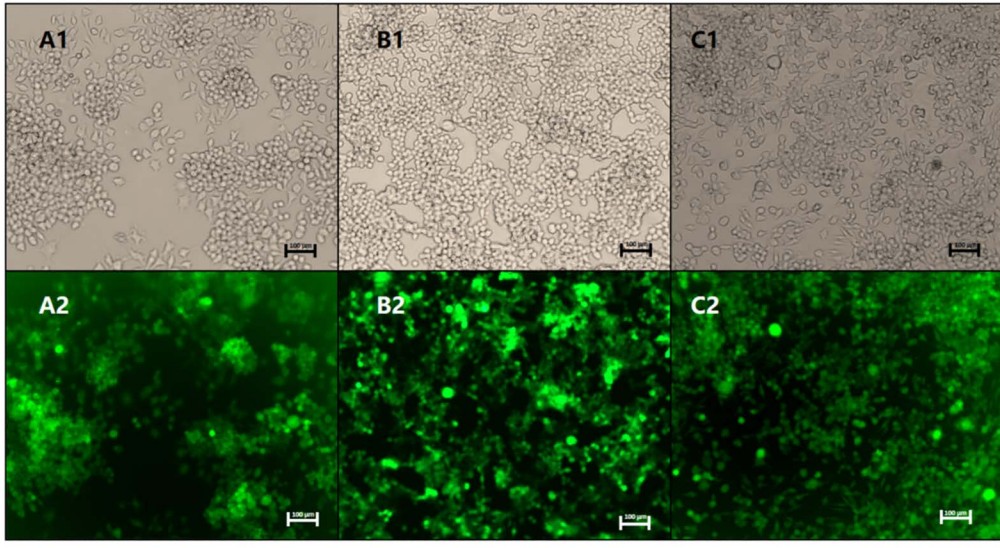

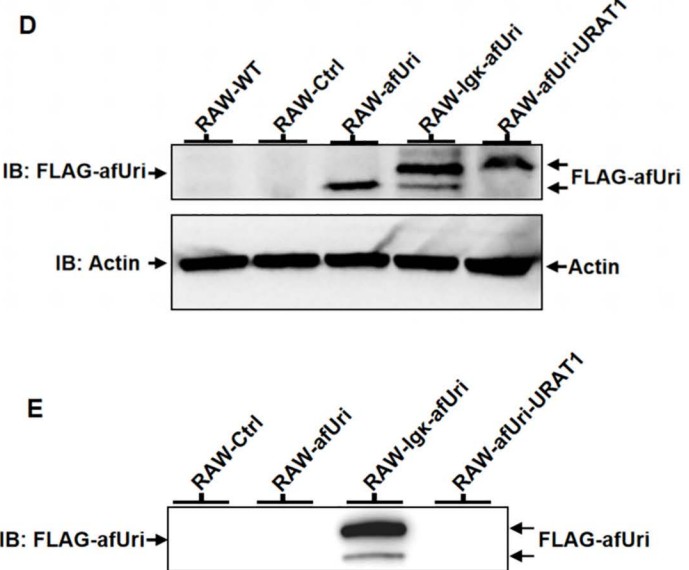

**Fig 2. (A1–C1) Bright-field images; (A2–C2) EGFP fluorescence images of cells after puromycin selection. (A)**: RAW-afUri-URAT1; **(B)**: RAW-Igκ-afUri; **(C)**: RAW-Ctrl. Scale bars = 100 μm. **(D)** Expression of intracellular FLAG-afUri. Lanes from left: RAW-WT, RAW-Ctrl, RAW-afUri, RAW-Igκ-afUri, RAW-afUri-URAT1. Arrow indicates FLAG-afUri. **(E)** Expression of FLAG-uricase in culture supernatants. Lanes: RAW-Ctrl, RAW-afUri, RAW-Igκ-afUri, RAW-afUri-URAT1.

To further clarify whether uricase is leaked or secreted in RAW-afUri-URAT1 cells, we detected FLAG-uricase in culture supernatants via western blotting. As shown in Fig 2E, specific bands were exclusively observed in the supernatant of the RAW-Igκ-afUri group. In contrast, no specific bands were detected in the supernatants of the RAW-afUri or RAW-afUri-URAT1 groups, despite their confirmed abundant intracellular uricase expression. These data demonstrate that the uricase expression pattern in the engineered RAW264.7 cells is consistent with the design.

## Engineered RAW264.7 cells show UA-lowering efficacy *in vitro*

We first evaluated the UA-lowering effect in culture medium. Different groups of engineered RAW264.7 cells were cultured and incubated with UA solutions as described in Materials and Methods. The UA concentrations of each group were quantified at 0, 24, 48, and 72 h. As shown in Fig 3, the RAW-afUri-URAT1 group exhibited a continuous and efficient UA-degrading trend: the supernatant UA concentration decreased gradually from an initial 556.0±37.0 µmol/L to 362.3±22.7 µmol/L at 24 h, 67.0±4.6 µmol/L at 48 h, and further to 0.7±0.6 µmol/L at 72 h, achieving almost complete degradation of UA within 72 h (Fig 3A, 3B). Notably, the RAW-Igκ-afUri group exhibited a more rapid UA-lowering activity. By 24 h post-incubation, the supernatant UA concentration dropped from 495.3±43.1 µmol/L to 1.0±1.7 µmol/L and remained at a near-zero level through 48 h 1.3±0.6 µmol/L and 72 h 2.3±1.5 µmol/L (Fig 3A, 3C). In contrast, the RAW-afUri group and RAW-Ctrl group showed no UA-degrading activity; instead, their supernatant UA concentrations slightly increased over time (Fig 3C, 3D). The data from RAW-afUri group suggested that the RAW264.7 cells with uricase expression are insufficient to degrade UA in the situations that the uricase cannot be secreted into the medium, or the UA is not transported into the cell. Taken together, our data showed that the engineered RAW264.7 cells could rapidly decrease UA level in medium.

To further clarify whether the UA-lowering effects of each group of RAW264.7 cell match their uricase expression pattern, we assessed UA-lowering effect of each supernatant. As shown in Fig 4, only the supernatants from RAW-Igκ-afUri group exhibited potent UA-degrading activity, with the UA concentration decreased sharply from the initial ~600 µmol/L to near undetectable levels by 24 h (Fig 4A), whereas supernatants from other groups showed no activity (Fig 4B-D). To further confirm that the UA uptake in RAW-afUri-URAT1 cells relies on URAT1, URAT1 inhibition assays were performed. As shown in Fig 4E, treatment with benzbromarone or Lesinurad abolished UA-lowering activity of RAW-afUri-URAT1 cells, strongly indicating that the degradation of UA in RAW-afUri-URAT1 cells takes place within the cell and relies on the transport of UA by URAT1.

## *In vivo* UA-lowering efficacy in hyperuricemic KM mice

The hyperuricemia model was established and the protocols were described in previous published work [35]. Briefly in this study, KM mice were randomly divided into 6 groups (n=8 per group), and the detailed treatment information for each group is summarized in S1 Table. The model was successfully established: at baseline (D0), sUA levels were comparable between the Mock group (162.01±20.82 µmol/L) and Model group (165.90±35.68 µmol/L, $P>0.05$); from D4 onwards, the Model group showed significantly higher sUA than the Mock group ($P<0.05$ or $P<0.01$), with sUA peaking at D8 (274.11±79.12 µmol/L) and then declining slightly until D16 (Fig 5A-C, S2 Table).

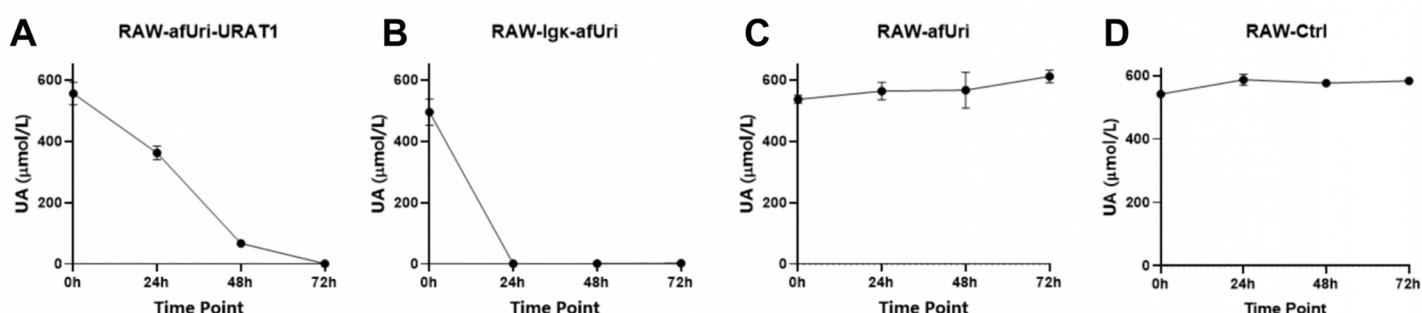

**Fig 3. *In vitro* UA-lowering efficacy of engineered macrophages.** Individual time-course curves of UA concentration for RAW-afUri-URAT1 **(A)**, RAW-Igκ-afUri **(B)**, RAW-afUri **(C)**, and RAW-Ctrl **(D)** groups, respectively. Data are presented as mean±SD (n=3).

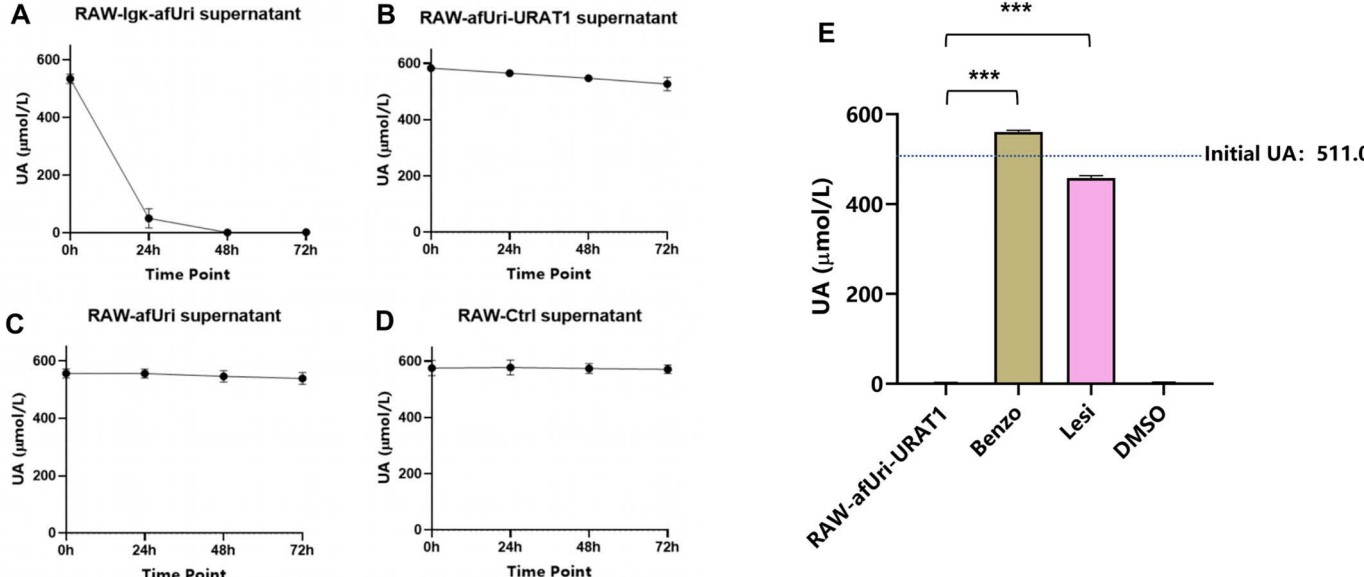

**Fig 4. UA-lowering mechanisms in each engineered macrophages. (A-D)** UA degradation in cell-free supernatants from each engineered cell group. **(E)** UA degradation in RAW-afUri-URAT1 cells in the presence of URAT1 inhibitors. Data are mean±SD. ***$P<0.001$.

As shown in Fig 5 D and E, RAW-Igκ-afUri and RAW-afUri-URAT1 successfully reduced sUA levels in hyperuricemic mice, with the intracellular "transport-degradation" type (RAW-afUri-URAT1) exhibiting superior overall efficacy. At baseline (D0), there was no significant difference in sUA levels between the two uricase-expressing groups and the RAW-Ctrl group ($P>0.05$). From D4 onwards, both groups showed significant sUA reduction compared with the RAW-Ctrl group (all $P<0.01$). The RAW-Igκ-afUri group exerted a faster initial UA-lowering effect (lower sUA at D4, $P<0.05$ vs. RAW-afUri-URAT1), while the RAW-afUri-URAT1 group gained prominent advantages in the middle and late stages (D12–D16), maintaining significantly lower sUA levels and achieving more durable efficacy.

In contrast, the sUA trend in the RAW-Ctrl group (Fig 5F, S4 Table) was consistent with that in the Model group (Fig 5C), with no statistically significant differences observed at all time points ($P>0.05$). This confirms that the UA-lowering effect was from the expression of uricase, rather than from the RAW264.7 macrophages themselves.

As a clinically approved uricase-based agent, rasburicase exhibited a rapid UA-lowering characteristic consistent with its pharmacodynamic profile (Fig 5G). At D4, its sUA level dropped sharply to 83.84±26.93 µmol/L and further decreased to 90.17±23.95 µmol/L at D12 after the second round of administration. However, rasburicase showed obvious efficacy rebound after each administration cycle: sUA increased by 54.0% from D4 to D8 and by 58.8% from D12 to D16. By the 16-day endpoint, its sUA level (143.19±38.21 µmol/L) was significantly higher than that of the RAW-afUri-URAT1 group (77.14±37.48 µmol/L, $P<0.01$) and RAW-Igκ-afUri group (103.52±32.09 µmol/L, $P<0.05$) (detailed statistical results summarized in S5 Table), failing to sustain the UA-lowering effect as the two engineered macrophage groups did.

### *In vivo* efficacy in liver uricase conditional knockout mice

In liver-specific uricase conditional knockout (Uox-CKO) mice (*Uox* flox/flox, *Alb Cre*+), serum uric acid (sUA) the serum uric acid level reached 179.5±20.1 µmol/L at day 28, which was significantly higher than that in wild-type mice (*Uox* flox/flox, *Alb Cre*-) (75.9±24.2 µmol/L, $P<0.001$), suggesting successful establishment of a genetic hyperuricemia model. Treatment

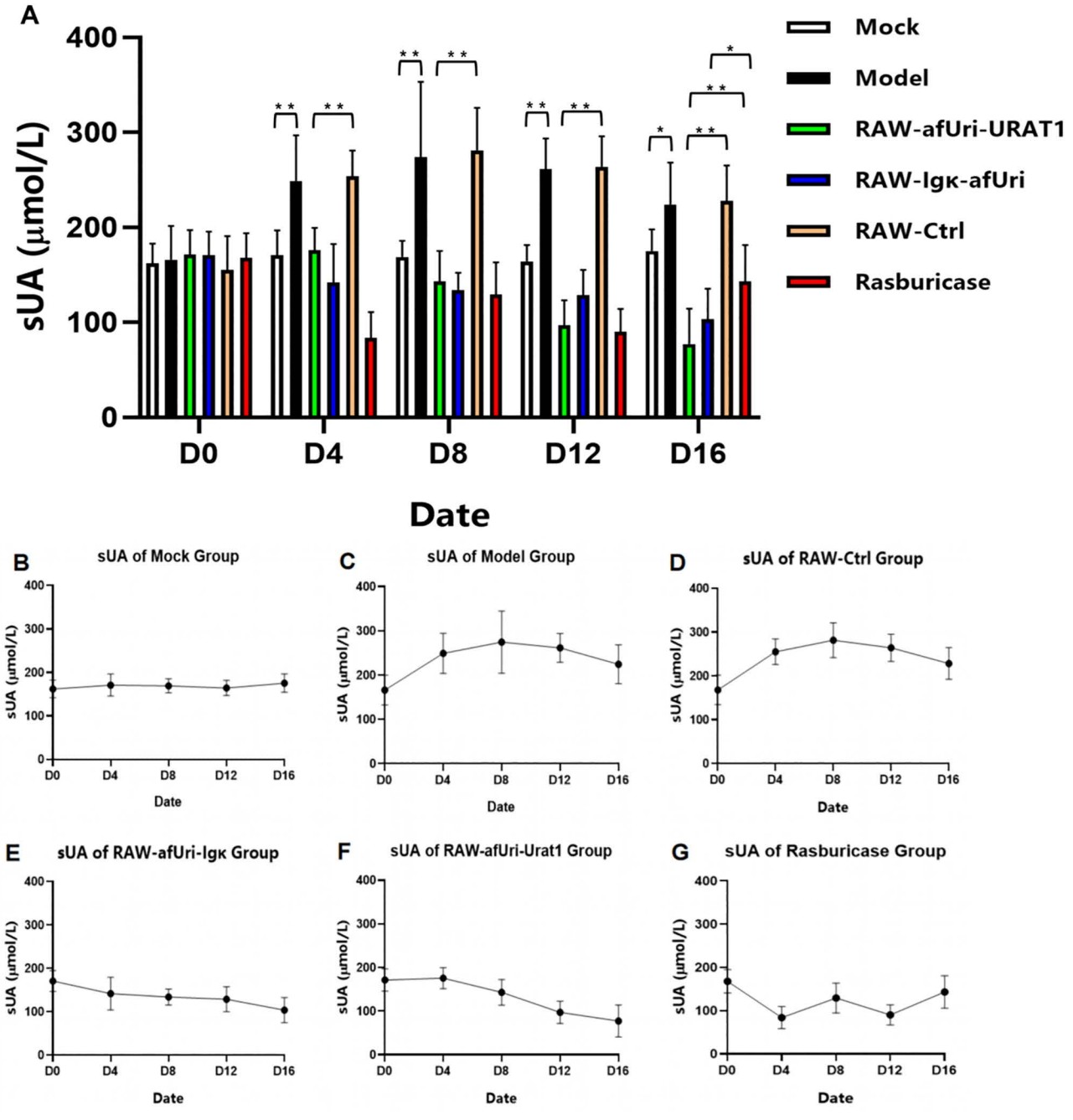

**Fig 5. Serum uric acid (sUA) levels in each group at different time points. (A)** Overview of sUA levels in all experimental groups (n = 8 per group). **(B–G)** Independent sUA profiles of Mock, Model, RAW-afUri-URAT1, RAW-Igκ-afUri, RAW-Ctrl, and Rasburicase groups, respectively. Data are presented as mean ± SD (analyzed by independent samples *t*-test).

with RAW-afUri-URAT1 significantly decreased sUA to 76.2 ± 15.9 μmol/L, nearly restoring levels to those of wild-type mice. In contrast, rasburicase only weakly lowered sUA to 142.4 ± 17.4 μmol/L, and RAW-Ctrl showed no obvious uric acid-lowering effect (167.7 ± 20.6 μmol/L). Notably, the urate-lowering effect of RAW-afUri-URAT1 was significantly

superior to both rasburicase and RAW-Igκ-afUri at the 28-day endpoint. These results demonstrated that RAW-afUri-URAT1 exerted durable and potent urate-lowering efficacy in a genuine genetic model of chronic hyperuricemia, complementing the transient dietary-induced KM mouse model and supporting the long-term potential of this cell-based strategy.

**Serum biochemistry and inflammatory factors in KM mice**

To evaluate the safety of engineered macrophages, we assessed biochemistry indicators and inflammatory makers in hyperuricemic mice and mice receiving cell treatment. All liver and renal function indicators (ALT, AST, TBIL, DBIL, CRE, UREA) remained within normal physiological ranges in all groups, with no significant differences among groups (S4 Fig), indicating good biosafety of engineered macrophages.

Levels of IL-1β, IL-6, and TNF-α were significantly elevated in all yeast extract-administered groups compared to the Mock control ($P < 0.001$), with the Model and RAW-Ctrl groups showing the most pronounced increases. While still higher than the Mock group ($P < 0.001$), the three treatment groups (RAW-afUri-URAT1, RAW-Igκ-afUri, and Rasburicase) exhibited markedly lower cytokine levels than the Model and RAW-Ctrl groups ($P < 0.001$). No significant differences were observed between the Model and RAW-Ctrl groups, nor among the three therapeutic groups. These results confirm that the engineered macrophages alleviate hyperuricemia-associated acute inflammation without inducing additional inflammatory responses (S5 Fig).

**Anti-uricase antibody responses**

Since afUri is heterologous protein to mice, we also need to evaluate the immunogenicity in the mice that received cell treatment. Anti-uricase antibody responses were determined by indirect ELISA using two coating antigens: commercial rasburicase and secreted *Aspergillus flavus* uricase (afUri) from RAW-Igκ-afUri supernatants. When coated with rasburicase, the RAW-afUri-URAT1 group exhibited an extremely low OD value ($0.62 \pm 0.07$), similar to the RAW-Ctrl group ($0.61 \pm 0.03$, $P > 0.05$), whereas the RAW-Igκ-afUri and Rasburicase groups showed markedly elevated antibody levels (both $P < 0.001$). When coated with afUri, RAW-afUri-URAT1 again displayed baseline OD values ($0.34 \pm 0.03$), comparable to RAW-Ctrl ($0.36 \pm 0.04$, $P > 0.05$), while RAW-Igκ-afUri generated robust antibody signals ($1.84 \pm 0.28$, $P < 0.001$). Notably, the Rasburicase group showed only weak cross-reactivity with afUri, indicating limited cross-recognition between the two antigens. Together, dual-ELISA results confirm that intracellularly confined uricase in RAW-afUri-URAT1 cells did not elicit detectable anti-uricase antibodies. In contrast, secreted uricase (RAW-Igκ-afUri) and free rasburicase were highly immunogenic, validating the low-immunogenicity advantage of the intracellular transport – degradation strategy (Fig 7).

## Discussion

Hyperuricemia and its complications pose a severe threat to public health, and the clinical unmet need for long-acting, low-immunogenicity UA-lowering therapies remains urgent. Uricase-based drugs, with their unique direct degradation mechanism, have become a breakthrough in refractory hyperuricemia treatment, but their application is limited by inherent immunogenicity. In this study we constructed a novel engineered macrophage delivery system co-expressing *A. flavus* uricase and URAT1, a UA transporter protein, to achieve uptake-intracellular degradation of environmental UA. This system exhibited superior UA-lowering efficacy and low immunogenicity in hyperuricemic mice, providing a new solution to the bottleneck of uricase clinical application.

The core innovation of this study lies in the ectopic co-expression of URAT1 and *A. flavus* uricase, which constructs a targeted "transport-degradation" cascade to overcome a longstanding limitation of uricase-based therapeutic delivery systems: the inefficient substrate accessibility that prevents encapsulated or immobilized enzymes from effectively engaging with circulating UA, hampering therapeutic potential. In contrast, engineered RAW264.7 macrophages in this study harness ectopically expressed URAT1—a highly specific UA transporter—to actively capture circulating UA and shuttle it directly into the cell internal compartment where uricase is also expressed. This design eliminates the substrate diffusion

barrier inherent to traditional approaches, ensuring an efficiently intracellular UA degradation. Data in Fig 3 and Fig 5 demonstrated that these macrophages armed with UA transporter and non-secretory uricase exhibit UA-lowering activity comparable to macrophages engineered to express a secretory one. This validates the feasibility and effectiveness of the transport-degradation strategy for UA-lowering, addressing a critical unmet need across uricase delivery platforms.

The RAW-Igκ-afUri group is designed as a critical positive control to verify whether macrophage-expressed uricase can achieve therapeutic effects comparable to clinical rasburicase, thus confirming the viability of the macrophage-based delivery strategy. Our *in vitro* data (Fig 3) clearly demonstrate that the RAW-Igκ-afUri group, which secretes uricase into the extracellular environment, achieved near-complete UA degradation within 24 h, with efficacy comparable to rasburicase. Although rasburicase acts more rapidly as a free enzyme, the secreted uricase from macrophages exhibited equally potent UA-lowering effects, confirming that macrophages can effectively express functional exogenous uricase. Notably, the RAW-afUri-URAT1 group achieved complete UA degradation (from $556.0 \pm 37.0$ µmol/L to $0.7 \pm 0.6$ µmol/L) within 72 h without extracellular secretion of uricase (Fig 2E). The cell-free supernatant assay (Fig 4) further validated its intracellular action mechanism: no uricase activity was detected in the supernatant of RAW-afUri-URAT1 cells, confirming that UA degradation is strictly intracellular and dependent on URAT1-mediated transport. More importantly, *in vivo* experiments (Fig 5, Fig 6) showed that in both acute and chronic hyperuricemia mouse models, both engineered macrophage groups maintained stable and persistent serum uric acid reduction for up to 28 days. In contrast, rasburicase exhibited obvious efficacy rebound over time. At the experimental endpoint, the macrophage-based treatment achieved significantly stronger and more durable urate-lowering effects than rasburicase, demonstrating that this delivery system not only matches but also surpasses rasburicase in both efficacy and durability.

Immunogenicity is the most critical limitation of uricase-based therapies, as exogenous uricase is recognized as a foreign antigen, inducing anti-drug antibodies (ADAs) [41], allergic reactions, and accelerated clearance [20], which severely restrict long-term clinical application. Strategies such as pegylation have been developed to reduce immunogenicity, but ADAs still develop in a substantial proportion of patients, leading to efficacy loss. In this study intracellular "transport-degradation" strategy offers a promising solution to this problem: uricase is sequestered within macrophages and not secreted into the extracellular environment (Fig 4), avoiding direct exposure to the immune system and thus reducing the

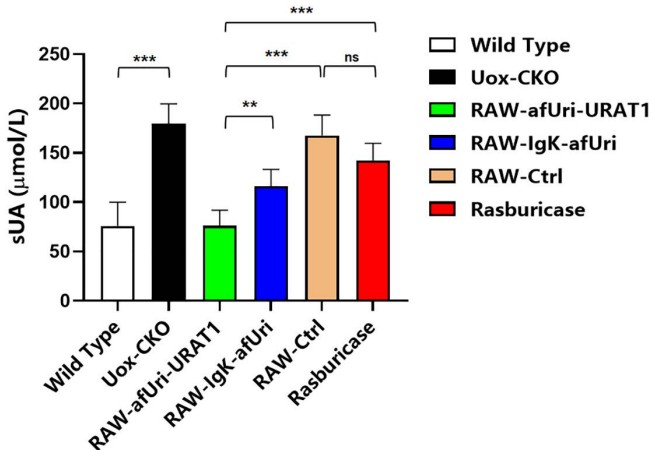

**Fig 6. RAW-afUri-URAT1 normalizes sUA in Uox-CKO mice at day 28.** Independent sUA levels of wildtype mice, Uox-CKO mice, Uox CKO mice treated with RAW-afUri-URAT1, RAW-Igκ-afUri, RAW-Ctrl, and Rasburicase groups, respectively.. Data are presented as mean $\pm$ SD. ***$P < 0.001$, **$P < 0.01$; ns, not significant.

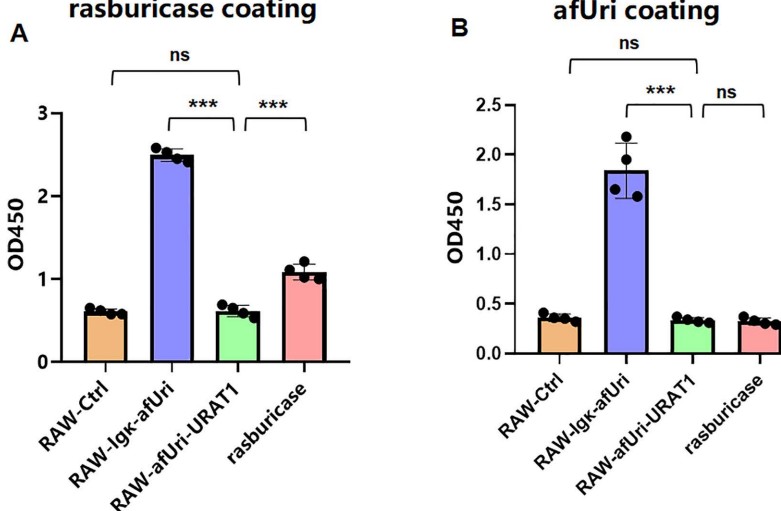

**Fig 7. Serum anti-uricase antibody detection by indirect ELISA. (A)** ELISA with rasburicase as the coating antigen. **(B)** ELISA with secreted afUri from RAW-Igκ-afUri cell supernatants as the coating antigen. Data are presented as mean ± SD (n = 4). *P* < 0.001; ns, not significant.

risk of ADA induction. Additionally, URAT1 is endogenously expressed in human renal proximal tubule cells, eliminating potential immunogenicity associated with the transporter itself. Consistent with the sustained efficacy without rebound, dual ELISA assays confirmed that RAW-afUri-URAT1 did not elicit detectable anti-uricase antibodies, and no excessive inflammatory responses were observed. These results directly validate the low immunogenicity and favorable safety profile of the intracellular "transport-degradation" strategy..

This study has several limitations that require further refinement. Firstly, although we employed two hyperuricemia mouse models to evaluate efficacy in both acute and chronic settings, the observation period was still relatively short for a comprehensive assessment of long-term safety and efficacy. Secondly, the study was conducted in a murine macrophage cell line (RAW264.7); future studies will need to be performed in primary autologous macrophages to better support clinical translation and personalized therapy. Thirdly, although previous studies have verified that intravenously administered RAW264.7 macrophages are predominantly distributed and accumulated in the liver [42], which is highly consistent with the liver-oriented metabolism of uric acid, the *in vivo* biodistribution of engineered macrophages in this study still need to be further determined.

## Conclusion

In this study, engineered RAW264.7 macrophages with ectopic co-expression of URAT1 and *A. flavus* uricase were developed as a novel cell-based delivery system for sustained UA degradation. The "transport-degradation" synergy is hypothesized to enable efficient uptake and intracellular breakdown of UA, while cellular encapsulation may shield the exogenous enzyme from immune recognition, thus reducing immunogenicity. *In vitro*, the engineered cells effectively degraded UA over 72 hours. In both acute and chronic hyperuricemic mouse models, RAW-afUri-URAT1 achieved potent and durable urate-lowering effects that were superior to rasburicase. No obvious organ toxicity, excessive inflammation, or anti-uricase antibody response was detected, supporting its favorable biosafety and low immunogenicity. This strategy combines high catalytic efficiency with enhanced biocompatibility and prolonged therapeutic action, addressing key limitations of current uricase therapies. These findings demonstrate that macrophage-mediated delivery offers a promising approach for treating refractory hyperuricemia with improved safety and durability, providing a foundation for future clinical translation of cell-based uricase therapies.

## Supporting information

**S1 Fig. Bright-field and EGFP images of RAW264.7 cells after lentiviral transfection.**
(TIF)

**S2 Fig. Flow cytometry analysis of engineered RAW264.7 cells post-transfection.**
(TIF)

**S3 Fig. Flow cytometry analysis of engineered RAW264.7 cells after puromycin selection.**
(TIF)

**S4 Fig. Serum biochemistry in KM mice.**
(TIF)

**S5 Fig. Serum levels of inflammatory cytokines in KM mice.**
(TIF)

**S1 Table. Interventions for hyperurecemic KM mice experiment.**
(TIF)

**S2 Table. Statistical comparison between Mock and Model groups.**
(TIF)

**S3 Table. Statistical comparison between therapeutic groups vs RAW-Ctrl group.**
(TIF)

**S4 Table. Statistical comparison between Model and RAW-Ctrl groups.**
(TIF)

**S5 Table. Pair-wise statistical comparison between therapeutic groups.**
(TIF)

**S1 File. Raw images.**
(PDF)

**S2 File. Metadata.**
(XLSX)

## Author contributions

**Data curation:** Qingyang Wang.

**Formal analysis:** Yupeng Li.

**Investigation:** Yuzhong Feng.

**Methodology:** Yuzhong Feng.

**Project administration:** Gang Liu.

**Resources:** Haolong Dong, Xianghua Xiong.

**Software:** Jiazhen Cui, Xuan Huang.

**Supervision:** Huipeng Chen.

**Validation:** Jiazhen Cui.

**Writing – original draft:** Yuzhong Feng.

**Writing – review & editing:** Gang Liu, Qingyang Wang.

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
