## [Decision Letter · Decision Letter 0]

24 Feb 2026

Dear Dr. Chen,

Thank you for submitting your manuscript to PLOS ONE. After careful consideration, we feel that it has merit but does not fully meet PLOS ONE’s publication criteria as it currently stands. Therefore, we invite you to submit a revised version of the manuscript that addresses the points raised during the review process.

We look forward to receiving your revised manuscript.

Kind regards,

Shengqian Sun

Academic Editor

PLOS One

Journal Requirements:

2. To comply with PLOS One submissions requirements, in your Methods section, please provide additional information regarding the experiments involving animals and ensure you have included details on (1) methods of sacrifice, (2) methods of anesthesia and/or analgesia, and (3) efforts to alleviate suffering.

Reviewers' comments:

Reviewer's Responses to Questions

**Comments to the Author**

1. Is the manuscript technically sound, and do the data support the conclusions?

Reviewer #1: Yes

Reviewer #2: Yes

2. Has the statistical analysis been performed appropriately and rigorously?

Reviewer #1: Yes

Reviewer #2: Yes

3. Have the authors made all data underlying the findings in their manuscript fully available?

Reviewer #1: Yes

Reviewer #2: Yes

4. Is the manuscript presented in an intelligible fashion and written in standard English?

Reviewer #1: Yes

Reviewer #2: Yes

Reviewer #1: 1. Lack of Direct Immunogenicity Assessment

The study claims reduced immunogenicity as a key advantage but provides no direct evidence (e.g., measurement of anti-drug antibodies, cytokine levels, or immune cell activation). Relying solely on sustained efficacy as an indirect proxy is insufficient and weakens the translational argument.

2. Inadequate Demonstration of URAT1 Functional Activity

While URAT1 expression is confirmed, no functional validation (e.g., UA uptake assays, URAT1 inhibition studies, or competitive transport experiments) is provided. Without this, the proposed "transport-degradation" mechanism remains speculative rather than proven.

3. Limited In Vivo Safety and Biodistribution Data

The study lacks essential preclinical safety data, including:

Biodistribution and persistence of engineered macrophages

Potential off-target effects or organ toxicity

Inflammatory or immune responses in tissues

This omission is significant for a cell-based therapeutic approach moving toward clinical translation.

4. Use of a Suboptimal Animal Model for Long-Term Evaluation

The yeast extract-induced hyperuricemia model is not ideal for durability studies because mice can endogenously upregulate uricase, leading to self-resolution of hyperuricemia. This confounds the interpretation of long-term efficacy and limits the validity of claims regarding sustained UA-lowering.

Reviewer #2: The authors have done a scientifically tremendous job and dive deeper in the scientific search to find the required output in a very technical and presentable manner. They have done relevant experiments with proper data output and presentation in the form of images and graphs. All details are perfect and scientific.

**Do you want your identity to be public for this peer review?** For information about this choice, including consent withdrawal, please see our For information about this choice, including consent withdrawal, please see our Privacy Policy .

Reviewer #1: **Yes:** Abiye BerihunAbiye Berihun

Reviewer #2: No

---

## [Author Response · Author response to Decision Letter 1]

30 Mar 2026

Dear Reviewers,

We greatly appreciate your thorough review and constructive comments, which have significantly helped us improve the quality and rigor of our manuscript. We would like to sincerely note that the experiments addressing immunogenicity, safety, URAT1 function, and chronic model validation had already been underway and continuously performed before and during the review period, and we have now fully integrated these data in the revised version.

Below is our point-by-point response to your comments:

Reply to Reviewer 1:

Comment 1: Lack of Direct Immunogenicity Assessment

The study claims reduced immunogenicity as a key advantage but provides no direct evidence (e.g., measurement of anti-drug antibodies, cytokine levels, or immune cell activation). Relying solely on sustained efficacy as an indirect proxy is insufficient and weakens the translational argument.

Response: We sincerely appreciate this critical comment. Given the lack of commercially available standardized ELISA kits for quantitative detection of anti-uricase antibodies, we established two semi-quantitative indirect ELISA assays using rasburicase and secreted A. flavus uricase (afUri) as capture antigens, respectively. Although semi-quantitative, these results clearly demonstrate that intracellularly confined uricase in RAW-afUri-URAT1, compared with secreted uricase, did not elicit detectable antibody responses in mice, validating low-immunogenicity advantage. The corresponding results have been added to the Results section as Fig 7.

Comment 2: Inadequate Demonstration of URAT1 Functional Activity.

While URAT1 expression is confirmed, no functional validation (e.g., UA uptake assays, URAT1 inhibition studies, or competitive transport experiments) is provided. Without this, the proposed "transport-degradation" mechanism remains speculative rather than proven.

Response: Thank you for this important suggestion. Standard uric acid uptake assays are difficult to perform in our system because intracellular uricase rapidly degrades transported uric acid, making direct uptake measurement unreliable. To overcome this limitation, we performed a URAT1 inhibition assay using benzbromarone and lesinurad. The results showed that pharmacological inhibition of URAT1 completely abolished uric acid degradation by RAW-afUri-URAT1 cells. This functional experiment indicates that URAT1mediated urate uptake is indispensable for the intracellular degradation mechanism, further validating the function of URAT1 in our engineered system. The data has been added to the Results section as Fig. 4E

Comment 3: Limited In Vivo Safety and Biodistribution Data.

The study lacks essential preclinical safety data, including:

Biodistribution and persistence of engineered macrophages

Potential off-target effects or organ toxicity

Inflammatory or immune responses in tissues

This omission is significant for a cell-based therapeutic approach moving toward clinical translation.

Response: We fully agree with your concern and have supplemented comprehensive safety data in the revision.

Safety assessment: We have added serum biochemistry (liver and renal function) and key inflammatory cytokines (IL-1β, IL-6, TNF-α) detection. All indicators remained within normal physiological ranges, and acute hyperuricemia-related inflammation was significantly alleviated, confirming good biosafety (S4 Fig, S5 Fig).

Biodistribution: The distribution of RAW264.7 cells in the body has been studied by other groups. Sapach et al. (2022, DOI: 10.1021/acsami.2c12004) have demonstrated that intravenously administered RAW264.7 cells were mainly accumulated in the liver under physiological conditions. We cited this paper and commented this issue in the Discussion section.

Comment 4: Use of a Suboptimal Animal Model for Long-Term Evaluation

Response: We greatly appreciate your professional comment. We also performed therapeutic experiments based on a liver uricase conditional knockout mice established by Pang et al. (2024, DOI: 10.1016/j.bbadis.2023.167009.). Unlike mice in gavage model, uricase liver-conditional knockout mice were unable to compensatory upregulate uricase expression as the gene has been knocked out. The data was not included in the initial manuscript because of not enough scale for animal experiments. During the under-review and revision periods, we have completed relative experiments. Our data showed that the urate-lowering efficacy by RAW-afUri-URAT1 sustained up to 28 days, suggesting good durability. We objectively note that 28 days still represents a limited observation period for lifelong clinical translation, and longer term evaluation will be included in our future studies. The data is presented as Fig. 6.

Reply to Reviewer 2:

We thank you for your kind and encouraging comments. We appreciate your recognition of the technical soundness, experimental rigor, and proper data presentation of our study. We will further refine the manuscript to align with the journal’s standards.

---

## [Decision Letter · Decision Letter 1]

6 Apr 2026

Ectopic Expression of Aspergillus flavus Uricase and URAT1 in Therapeutic Cells Promotes Intracellular Degradation of Uric Acid in Hyperuricemic Mice

PONE-D-26-03089R1

Dear Dr. Chen,

We’re pleased to inform you that your manuscript has been judged scientifically suitable for publication and will be formally accepted for publication once it meets all outstanding technical requirements.

Kind regards,

Shengqian Sun

Academic Editor

PLOS One

Additional Editor Comments (optional):

Reviewers' comments:

Reviewer's Responses to Questions

**Comments to the Author**

Reviewer #1: All comments have been addressed

2. Is the manuscript technically sound, and do the data support the conclusions?

Reviewer #1: Yes

3. Has the statistical analysis been performed appropriately and rigorously?

Reviewer #1: Yes

4. Have the authors made all data underlying the findings in their manuscript fully available?

Reviewer #1: Yes

5. Is the manuscript presented in an intelligible fashion and written in standard English?

Reviewer #1: Yes

Reviewer #1: (No Response)

**Do you want your identity to be public for this peer review?** For information about this choice, including consent withdrawal, please see our For information about this choice, including consent withdrawal, please see our Privacy Policy .

Reviewer #1: No

---

## [Editor Report · Acceptance letter]

PONE-D-26-03089R1

PLOS One

Dear Dr. Chen,

I'm pleased to inform you that your manuscript has been deemed suitable for publication in PLOS One. Congratulations! Your manuscript is now being handed over to our production team.

Kind regards,

on behalf of

Dr. Shengqian Sun

Academic Editor

PLOS One